# Genome-Wide Identification and Characterization of Effector Candidates with Conserved Motif in *Falciphora oryzae*

**DOI:** 10.3390/ijms25010650

**Published:** 2024-01-04

**Authors:** Mengdi Dai, Zhenzhu Su, Xueming Zhu, Lin Li, Ziran Ye, Xiangfeng Tan, Dedong Kong, Xiaohong Liu, Fucheng Lin

**Affiliations:** 1State Key Laboratory for Managing Biotic and Chemical Treats to the Quality and Safety of Agro-Products, Institute of Plant Protection and Microbiology, Zhejiang Academy of Agricultural Sciences, Hangzhou 310021, China; daimd@zaas.ac.cn (M.D.); zhuxm@zaas.ac.cn (X.Z.); lilin@zaas.ac.cn (L.L.); 2State Key Laboratory for Managing Biotic and Chemical Treats to the Quality and Safety of Agro-Products, Institute of Digital Agriculture, Zhejiang Academy of Agricultural Sciences, Hangzhou 310021, China; yezr@zaas.ac.cn (Z.Y.); tanxf@zaas.ac.cn (X.T.); kongdd@zaas.ac.cn (D.K.); 3State Key Laboratory for Managing Biotic and Chemical Treats to the Quality and Safety of Agro-Products, Institute of Biotechnology, Zhejiang University, Hangzhou 310058, China; zzsu@zju.edu.cn (Z.S.); xhliu@zju.edu.cn (X.L.); 4Xianghu Laboratory, Hangzhou 311231, China

**Keywords:** genome-wide, effector, conserved motif, endophyte

## Abstract

Microbes employ effectors to disrupt immune responses and promote host colonization. Conserved motifs including RXLR, LFLAK-HVLVxxP (CRN), Y/F/WxC, CFEM, LysM, Chitin-bind, DPBB_1 (PNPi), and Cutinase have been discovered to play crucial roles in the functioning of effectors in filamentous fungi. Nevertheless, little is known about effectors with conserved motifs in endophytes. This research aims to discover the effector genes with conserved motifs in the genome of rice endophyte *Falciphora oryzae*. SignalP identified a total of 622 secreted proteins, out of which 227 were predicted as effector candidates by EffectorP. By utilizing HMM features, we discovered a total of 169 effector candidates with conserved motifs and three novel motifs. Effector candidates containing LysM, CFEM, DPBB_1, Cutinase, and Chitin_bind domains were conserved across species. In the transient expression assay, it was observed that one CFEM and one LysM activated cell death in tobacco leaves. Moreover, two CFEM and one Chitin_bind inhibited cell death induced by Bax protein. At various points during the infection, the genes’ expression levels were increased. These results will help to identify functional effector proteins involving omics methods using new bioinformatics tools, thus providing a basis for the study of symbiosis mechanisms.

## 1. Introduction

Interactions between plants and microbes could be pathogenic or mutualistic, influencing the growth, immunity, and tolerance to abiotic stresses of plants [1,2]. The advantageous mutualism relationship, which supplies plants with growth-restricting elements such as nitrogen and phosphorus, holds great significance in the field of agriculture as it diminishes the requirement for fertilizers. Yuan et al. discovered *Falciphora oryzae*, a non-pathogenic dark septate endophyte, in the roots of wild rice in 2007 [3]. Previous research has indicated that *F. oryzae* enhanced the development and biomass of rice crops, evident in the augmented growth of plant shoots, stem diameter, leaf width, fresh weight, and chlorophyll content. Simultaneously, there was an increase in the levels of protective enzymes, H_2_O_2_ amount, and disease-resistant gene expression to enhance the systemic resistance when *F. oryzae* colonized the roots of rice [4]. Despite the available data indicating a symbiotic relationship between rice and *F. oryzae*, the precise molecular mechanism by which *F. oryzae* inhibits the host’s defense response and forms a symbiotic relationship remains unknown.

Effector proteins play an important role in the interaction between fungi and plants. Pathogens employ effectors to resist host defense response to complete their life cycles [5]. Endophytic fungi also secrete effectors to host cells to establish symbiotic relationships and regulate host defense [6]. Therefore, the identification of endophytic fungi effectors is a major topic for exploring the mechanism of endophyte–plant interaction. Effectors contain specific motifs that target proteins in the host, resulting in unique impacts on infection, virulence, and various physiological and molecular activities in plants. The conserved motif RXLR, which contained a preserved Arg-X-Leu-Arg sequence situated in the N-terminal areas, was discovered in both oomycetes and fungi. The RXLR motif is frequently succeeded by a dEER pattern around 20 to 25 amino acids further down [7]. RxLR effectors are widely found in Phytophthora species, and they regulate plant immunity through different molecular mechanisms [8]. The CRN (for CRinkling and Necrosis) motif, like RLXR, also appears in the effector proteins of multiple kingdoms. More than 200 genes were found in *Phytophthora infestans* and *Phytophthora sojae* [9]. After gene sequencing, few CRN genes were discovered in mycorrhizal and endophytic fungi and their functions in symbiosis remained unknown [10]. Carbohydrate-binding domains known as LysM motifs are extensively preserved across various fungal kingdoms. Slp1, one of the LysM effectors in *M. oryzae*, interacted with chitin to inhibit chitin-induced host defense and played a crucial role in tissue invasion and the expansion of disease lesions [11]. The interaction between the mutualistic fungi and plants is also significantly influenced by LysM effectors. Zuccaro et al. discovered several effector proteins in *P. indica*, including PIIN_00867, PIIN_08721, and PIIN_08723, all of which contain the LysM motif [10]. CFEM motifs were commonly found in the N-terminus of extracellular membrane proteins in fungi and consisted of eight cysteine residues in the eight-Cys-containing domain [12]. In *M. oryzae*, the characteristic CFEM proteins are ACI1 and Pth11. Structure analysis revealed that Pth11 had a significant impact on the control of redox and the differentiation of appressorium [13]. ACI1 controlled appressorium development and the infection process [14]. As for endophytic fungi, there were three proteins containing CFEM motifs found in *P. indica*, called PIIN_03540, PIIN_05622, and PIIN_08499 [15]. While these conserved sequence motifs are well-established methods for predicting certain types of effectors in pathogens, effector proteins containing conserved motifs have not been identified and functionally analyzed in many endophytes.

Genome sequencing provides insights into the repertoires of effector candidates in highly specialized pathogens based on the identification of secreted proteins expressed during biotrophic growth. In *Sporisorium scitamineum*, 68 secreted effector candidates were identified in the genome and were predicted to be involved in defense response [16]. Based on genome sequences and transcriptome analysis, a planta-induced candidate effector repertoire was generated in *Sclerospora graminicola*. Of the 1,220 genes that encode presumed secreted proteins, 91 genes significantly altered their expression levels during plant tissue infection [17]. In recent years, several genomic sequences for endophytes have been released; however, effector candidate repertoires with conserved motifs have not been established. Identification and characterization of effector candidates based on sequence characteristics and conserved motifs will be an important challenge in endophytes.

It was clear that *F. oryzae* and rice could form a mutualistic relationship, but the mechanism of their interaction was unknown, and its effector repertoires remained scarce. Furthermore, there was a lack of reports on the role of effectors in *F. oryzae* and their expression data. Therefore, this study aimed to identify and characterize *F. oryzae* effector candidates through data mining and bioinformatics tools, and verify them by qRT-PCR. The findings will provide a basis for future studies on the interaction between plants and endophytes in terms of symbiosis and immunity.

## 2. Results

### 2.1. Genome-Wide Identification of Effector Candidates

Genome sequences of *F. oryzae* were downloaded from the NCBI genomic database (Genebank accession number: GCA_000733355.1). We obtained the gene annotation by Swiss-port. Signal peptides play a role in directing protein secretion in the process of interacting with plants. Using SignalP, 622 proteins containing signal peptides were screened out as secreted proteins. These secreted proteins were detected by EffectorP to evaluate the effector probability, and a total of 227 proteins were screened out as effector candidates (workflow presented in Figure 1a). 

### 2.2. Characterization of Effector Candidates Containing RXLR-like, CRN-like, and Y/F/WxC Motifs

By utilizing hmmsearch, we obtained the RXLR motif (PF16810) and a regular code for the RXLR-like motif ([RKH]X[LIMFYWK][RALQGTF]) mentioned in a prior study [7]. Through this process, we successfully identified 169 potential RXLR-like effectors (Figure 1b). Out of all of the candidates, only a single RXLR-like effector had an identical RXLR sequence (Appendix A). By utilizing a regex code of dEER-motif ([ED][ED][KR]) obtained from a prior study [7], a total of 28 potential RXLR-dEER-like effectors were discovered within the RXLR-like proteins (Appendix A). Weblog was utilized to display hidden Markov models of conserved regions for a total of 169 RXLR-like and 28 RXLR-deeR-like effector candidates (Figure 2a).

Using hmmsearch and regex reported in a previous study, we also identified 18 Y/F/WxC effector candidates (Appendix A). Weblog was used to visualize the hidden Markov models of 18 Y/F/WxC effector candidates’ conserved regions (Figure 2b). Nevertheless, there were no identified secreted proteins exhibiting the [SG]-P-C-[KR]-P and LFLAK (CRN) motifs.

### 2.3. Characterization of Effector Candidates Containing LysM, CFEM, Cutinase, DPBB_1, and Chitin_bind Domains

Other effector candidates with conserved domains, such as LysM, CFEM, Cutinase, DPBB_ 1, and Chitin bind domains, were identified according to the workflow presented in Figure 1b. The hidden Markov models for the conserved domains of LysM (PF01476), CFEM (PF05730), DPBB_1 (PF03330), Cutinase (PF01083), Chitin_bind_1 (PF00187), EAR (PF07897), and ToxA (PF11584) were obtained directly from the Pfam website. Using hmmsearch, two LysM, three CFEM, three DPBB_1, three Cutinase, and one Chitin_bind were screened out as effector candidates (Table 1). No secreted proteins with conserved domains of ToxinA and EAR were identified.

Through phylogenetic analysis, we found that the effector candidates containing LysM, CFEM, DPBB_1, Cutinase, and Chitin_bind domains were conserved across species. The closest homologs of these conserved domains were *Magnaporthiopsis poae* and *Gaeumannomyces tritic* (Figure 3). Furthermore, the conservation of CFEM motif sequences, especially the eight cysteine residues, was demonstrated by multiple sequence alignment and Weblog (Figure 4). To characterize other consensus patterns typical for *F. oryzae*, amino acid sequences encoded by the LysM, DPBB_1, and Cutinase genes from the *F. oryzae* genome were aligned (Appendix A). 

Using GSCS, the codon sequences of effector candidates were compared with their DNA sequences to determine the location and quantity of introns and exons in each gene. As a result, we found that most effector candidates had two to four exons for protein translation (Figure 5).

Using MEME, the motifs of 58 effector candidates with no conservative structure were detected. The three most statistically significant (low E-value) novel motifs were found (Figure 6a), with motifs one to three appearing in two, three, and two effector candidates, respectively. Three motifs can be seen in the effector candidates g8761 and g10328 (Figure 6b).

### 2.4. Effector Candidates Induced/Suppressed Cell Death in N. benthamiana Leaves

Effector candidates with conserved domains (LysM, CFEM, Cutinase, DPBB_1, and Chitin_bind) were constructed as transient expression vectors and introduced into *N. benthamiana* leaves through *A. tumefaciens* infiltration, with Bax serving as a positive control. From the results, we found that the effector candidates of g11649 (containing CFEM domain) and g3960 (containing LysM domain) caused cell death in tobacco leaves (Figure 7a). In comparison to the necrosis-inhibiting capacity of GFP (negative control), g12356 (containing Chitin bind domain), g10237, and g11871 (containing CFEM domain) could suppress Bax-induced cell death when co-infiltrated with Bax (Figure 7b). Nevertheless, no effector candidates containing conserved Cutinase and DPBB_1 domains were found to induce or suppress cell death in tobacco leaves.

### 2.5. The Expression Characteristics of Effector Candidates

In the transient expression assay, we have found five effector candidates that could induce/suppress cell death in *N. benthamiana* leaves. In this study, we analyzed these five effector candidates in the stage of *F. oryzae* infection on rice roots through qRT-PCR. From the results, we found that the expression level of *g3960* was upregulated during 12–36 dpi, then decreased, and showed the highest expression at 60 dpi. *g11649* peaked at 12 dpi, followed by downregulation, and was upregulated during 24–60 dpi. *g10237*, *g11871*, and *g12356* showed the highest expression at 72 dpi. After upregulating at 36 dpi, *g10237* dropped at 48 dpi, then climbed significantly between 48 and 72 dpi. *g11871* was upregulated at 24 dpi, then decreased during 24–48 dpi. *g12356* was expressed highly at 12 dpi, and the expression was downregulated at 24 dpi, followed by a wave trend during 36–60 dpi, peaking at 72 dpi (Figure 8). The findings indicated that these genes were activated during interaction with the host and were upregulated at various infection stages.

## 3. Discussion

Effectors are important factors in the interaction between plants and microbes. Microbes release effectors to suppress immune responses and facilitate host colonization [18]. Some fungal effectors have certain characteristics, such as the conserved motifs RxLR in oomycetes and the sequence similarity of [Y/F/W]xC in powdery mildew. However, most fungal effectors do not typically exhibit highly conserved amino acid sequences. Therefore, the prediction of effectors is mainly dependent on bioinformatics. Previous studies have demonstrated that this method can predict the effector proteins of fungi [19]. The first step of effector protein prediction is usually to forecast extracellular secretion signals. Based on the SP site, SignalP V6.0 employs a range of artificial neural networks to forecast peptides as either signals or non-signals. The main role of signal peptides in the interaction with plants is to guide protein secretion. The specific location of the signal peptide was predicted by SignalP v5.0 software, which further increased the possibility of it being an effector protein. In the current study, a total of 622 proteins were predicted to be secreted proteins by SignalP. Further, we used EffectorP to predict whether they were effector proteins. EffectorP, a bioinformatics program, employs machine learning techniques to forecast potential effectors based on protein properties [20]. From the prediction program, 227 proteins were identified as potential effectors. It is common to use bioinformatics tools to screen effector proteins; however, molecular biological methods are still necessary, such as transgenic technology, and host-induced gene silencing (HIGS) to verify the function of effector proteins.

Effectors containing conserved motifs like RXLR, LFLAK-HVLVxxP (CRN), Y/F/WxC, CFEM, LysM, Chitin_bind, DPBB_1 (PNPi), and Cutinase are commonly referred to as “core effectors”, functioning in pathogenicity/symbiosis mechanisms and plant defense strategies [21]. Some fungal effectors have certain characteristics, such as conservative motifs RXLR in oomycetes. *Phytophthora capsici*, an oomycete, possesses 400 RxLR motifs that play a role in the virulence of the pathogen [22]. In contrast, biotrophic fungi like *Albugo candida* and *Albugo laibachii* have a smaller number of effectors containing RXLR or RXLR-like motifs [23]. Moreover, for necrotrophic pathogens, there are only two RXLR motifs found in *Pythium ultimum* and none in *Aphanomyces euteiches* [24,25]. However, Mycorrhiza and endophytic fungi have received less attention compared to the significance of the RXLR motif in pathogens. There were 167 effectors with RXLR-EER found in *Piriformospora indica*. The hypothesis was raised that the RXLR motif could enable *P. indica* to evade the immune defense system of host plants, due to its extensive host range and strong colonization ability [10]. In this study, we identified 169 RXLR-like effector candidates in endophytes *F. oryzae*. Among them, only one RXLR-like effector candidate contained an identical RXLR sequence (Appendix A). By employing a regex code of dEER-motif ([ED][ED][KR]), we identified 28 RXLR-dEER-like effector candidates from the RXLR-like proteins (Appendix A). These results supported the hypothesis that endophytes might escape host defense responses through RXLR motifs.

In wheat stem rust (*Puccinia graminis*), wheat leaf rust fungus (*Puccinia triticina*), and barley powdery mildew (*Blumeria graminis*), Y/F/WxC motifs are present in the N-terminal regions of effectors [26,27]. In barley powdery mildew, effector proteins containing Y/F/WxC motifs contribute to pathogenic virulence [28]. In our study, we screened out 18 effector candidates in *F. oryzae* that contained Y/F/WxC motifs (Appendix A). However, none of the 18 identified effector candidates were found to promote or inhibit plant cell death (Figure 7). The detailed functions of the interaction with rice needs further exploration.

The domains CFEM, LysM, DPBB_1, Chitin_bind, and Cutinase are characteristic of effectors. The effector proteins with the CFEM domain were involved in plant immunity [29]. In *P. triticina*, the CFEM effector PTTG_08198 accelerated cell death triggered by Bax in tobacco leaves [27]. The BcCFEM1 protein containing the CFEM motif was significantly upregulated in the early stage of infection of *Phaseolus vulgaris* leaves and induced cell death in tobacco leaves. The deletion of *BcCFEM1* showed significant impacts on virulence, conidial production, and stress tolerance. In addition, the protein has a putative GPI anchoring site at the C-terminal, which affects effectors’ secretion and translocation. Therefore, it plays a role in the early stages of the *Botrytis cinerea* infection [30]. In our study, we have identified three CFEM effector candidates (Table 1). Among them, one CFEM effector candidate caused cell death in tobacco leaves and was downregulated at the early infection stage. The other two CFEM effector candidates inhibited Bax-induced cell death and were upregulated at the early infection stage (Figure 7 and Figure 8). These results suggested that CFEM motifs were directly related to the endophytes–plant immune response. The expression of different CFEM effector proteins at various infection stages might facilitate the colonization of endophytes in roots.

In pathogens, LysM effectors are typically invaded by suppressing the chitin-triggered immunity in host plants [31]. VdLs17, as an effector containing the LysM domain, inhibited chitin-induced immune response by binding to chitin, thus protecting fungal mycelia from plant hydrolase hydrolysis [32]. In addition, LysM effectors have a significant function in the symbiotic relationship between endophytes and plants [33]. Some endophytes secrete LysM effectors that bind to chitin on the cell wall, masking the fungus to avoid degradation by plant chitinase [34]. Suarez-Fernandez et al. indicated that LysM domains play a key role in the ability of endophytes to colonize plants [35]. In this study, we have identified two LysM effector candidates (Table 1). Transient expression assay showed that the effector candidate g3960 (containing LysM domain) demonstrated induction of cell death in tobacco leaves (Figure 7a). In addition, the LysM effector candidate was upregulated at the early infection stage (Figure 8). This result is different from that of the LysM effectors of the pathogen to inhibit the chitin response in plants. *F. oryzae* released LysM effectors to bind to chitin in the plant cell walls in the early stage of infection, escaping the degradation of chitinase, thus helping fungi colonize rice successfully. However, there are still some questions about LysM effectors: (1) What is the subsequent reaction after the LysM effector escapes chitinase hydrolysis in the host? (2) Is there any involvement of CFEM, RXLR, and other effectors mentioned above? These molecular mechanisms of effectors in endophytes need to be further explored.

## 4. Materials and Methods

### 4.1. Plant Materials and Fungal Strains

The strain A20-1-1 of *F. oryzae*, a fungal endophyte, was grown in the absence of light at 25 °C on potato dextrose agar (PDA, Difco, Detroit, MI, USA). In a greenhouse, the rice variety CO-39 and tobacco were cultivated under conditions of 16 h of light and 8 h of darkness, with temperatures ranging from 25 to 22 °C. The *Escherichia coli* strain DH5α was utilized for plasmid extraction, while the *A. tumefaciens* strain GV3101 was employed for transient expression.

### 4.2. Genome-Wide Identification of Effector Candidates in F. oryzae

In order to identify effector candidates, the whole-genome sequence of *F.oryzae* was acquired from the NCBI database (https://www.ncbi.nlm.nih.gov/datasets/genome/GCA_000733355.1/ accessed on 23 Octorber 2023). Swiss-port was utilized to compare the genome sequence with the database sequence for gene annotation. To forecast the signal peptides at the N-terminal, SignalP (https://services.healthtech.dtu.dk/service.php?SignalP-5.0 accessed on 30 October 2023) was utilized. The methods for submitting and predicting sequences were carried out in accordance with the explanation provided by Nielsen [36]. Proteins with a D score of YES were selected as secreted proteins. After extraction, the selected sequences were sent to EffectorP (https://effectorp.csiro.au/ accessed on 2 November 2023), as stated by Sperschneider [37]. An assessment was carried out to determine the likelihood of the effectors, which is determined by factors such as the weight of molecules, the charge number, and the characteristics of cysteine, serine, and tryptophan in the amino acid sequences. The neighbor protein alignment was matched using Clustal X 2.1. Mega version X was used for the construction of phylogenetic trees [38].

### 4.3. Structural Analysis of Effector Candidates

The inferred proteins were conducted through hmmsearch, regex, and homology analysis. The conserved domains in each effector candidate were predicted using Pfam (http://pfam.xfam.org/ accessed on 17 November 2023). Common motifs such as RXLR, DPBB_1 (PNPi), Y/F/WxC, CFEM, LysM, EAR, LFLAK (CRN), [SG]-P-C-[KR]-P, ToxA, EAR, Cutinase, and Chitin_bind were employed based on a personalized procedure (Figure 1). The HMM models for RXLR (PF16810), CFEM (PF05730), LysM (PF01476), EAR (PF07897), ToxA (PF11584), DPBB_1 (PF03330), Cutinase (PF01083), and Chitin_bind (PF00187) were obtained from the Pfam site. For RXLR-like motifs, the modified regex code of (/w [10,40}\w [1,96][RKH]/w[LIMFYWK][RALQGTF]) was used from the protein database [39]. Proteins containing a complete RXLR-deeR-like motif were identified by further screening the gathered RXLR-like effector candidates using the regex code [ED][ED][KR] from a previous study [39]. Furthermore, the regex codes of (^\w{10,40}\w{1,30}[YFW]\wC), (^\w{10,40}\w{1,96}L[FYRL][LKF][ATVRK][KRN]), and (^\w{10,40}\w{1,96}[GS]PC[KR]P) were used to screen the effector candidates containing motifs of Y/F/WxC, LFLAK (CRN) and [SG]-P-C-[KR]-P. The hidden Markov models were visualized using Weblog (http://weblogo.berkeley.edu/ accessed on 17 November 2023). The effector candidates without conservative structure were detected by MEME v5.3.2 software.

### 4.4. Transient Expression in N. benthamiana

Effector candidates with conserved domains were amplified from the wild-type strain’s cDNA and then cloned into a tobacco rattle virus (TRV) vector (Appendix A). These clones were used to create transient expression vectors in *N. benthamiana*. The TRV effector vectors were introduced into the *A. tumefaciens* strain GV3101 through the freeze/thaw method, as described by Wise [40]. After being subjected to centrifugation, the agrobacterium samples were suspended in 10 mM MgCl_2_ to OD_600_ = 0.2. Following a 3 h incubation period without light, the vectors were injected into 3-week-old *N. benthamiana* leaves. A cell death phenotype was observed in GFP control samples around 3 to 5 days following Bax inoculation. If the effector candidates were injected without any symptoms, they would be co-infiltrated with Bax. Initially, the leaves were infiltrated with *A. tumefaciens* cells carrying effector candidates, and 24 h later, Bax was also infiltrated at the same location. The analysis was conducted on three separate occasions for each sample. The test was conducted a minimum of two times.

### 4.5. Isolation of RNA and Gene Expression Level Assay by qRT-PCR

To examine the gene expression in the initial phase of infection, rice roots inoculated with *F.oryzae* were collected at 12, 24, 36, 48, 60, and 72 h post-inoculation (hpi). Trizol reagent (Invitrogen, Carlsbad, CA, USA) was used to extract the samples according to the guidelines provided by the manufacturer. Reverse transcriptase (Takara, Japan) was utilized to produce the initial cDNA from the isolated total RNA. The qRT-PCR experiments were conducted with a Mastercycler ep realplex Thermal Cycler (Eppendorf, Westbury, NY, USA), involving two stages and a subsequent analysis of the melting curve. For calculating relative transcription levels, the tubulin gene (FAOR_9422) was employed as an endogenous reference. Primers for five chosen genes that caused cell death in *N. benthamiana* were designed using qRT-PCR (Appendix A). The 2^−ΔCt^ method was employed to determine the relative levels of gene expression [41]. Each treatment was replicated three times, with the experiments being repeated at least twice.

## 5. Conclusions

Through genome-wide characterization, a total of 227 effector candidates were screened out by SingalP and EffectorP. In addition to three newly discovered motifs, various recognized motifs including RXLR, LFLAK-HVLVxxP (CRN), Y/F/WxC, CFEM, LysM, Chitin-bind, DPBB_1 (PNPi), and Cutinase were detected. The effector candidates with conserved domains showed high similarity to *Magnaporthiopsis poae* and *Gaeumannomyces tritic*. In the transient expression assay, we discovered two effector candidates that triggered cell demise and three effector candidates that suppressed cell demise in tobacco leaves. At various points during the infection, the genes’ expression levels were increased. This is the first time that RXLR, LFLAK-HVLVxxP (CRN), Y/F/WxC, CFEM, LysM, Chitin-bind, DPBB_1 (PNPi), and Cutinase effector proteins have been reported in the *F. oryzae* genome. Further research on the interaction between endophytic effector proteins and host plants will provide a new research field for confirming the symbiosis mechanism between endophytes and host plants.

## Figures and Tables

**Figure 1 ijms-25-00650-f001:**
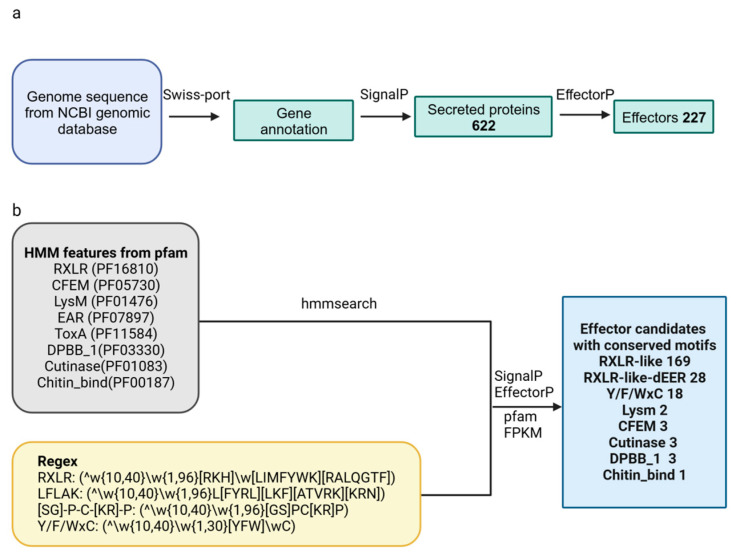
Workflow of identifying effector candidates. (**a**) The genome sequences were downloaded from the NCBI genomic database and annotated by Swiss-port. A total of 622 secreted proteins were detected by employing SingalP. EffectorP was used to identify these secreted proteins to assess their likelihood of being effectors. A total of 227 proteins were identified as effector candidates. (**b**) Using hmmsearch, we obtained the RXLR motif (PF16810) and a regex code for an RXLR-like motif. A total of 169 effector candidates with RXLR-like motifs were discovered. A total of 28 RXLR-dEER-like effector candidates were found among the RXLR-like proteins using a regex code of dEER-motif. Additionally, hmmsearch and regex were able to identify 18 Y/F/WxC. Effector candidates LysM, CFEM, DPBB_1, Cutinase, and Chitin_bind were identified using the downloaded hmm models for the conserved domains of LysM (PF01476), CFEM (PF05730), DPBB_1 (PF03330), Cutinase (PF01083), Chitin_bind (PF00187), EAR (PF07897), and ToxA (PF11584).

**Figure 2 ijms-25-00650-f002:**
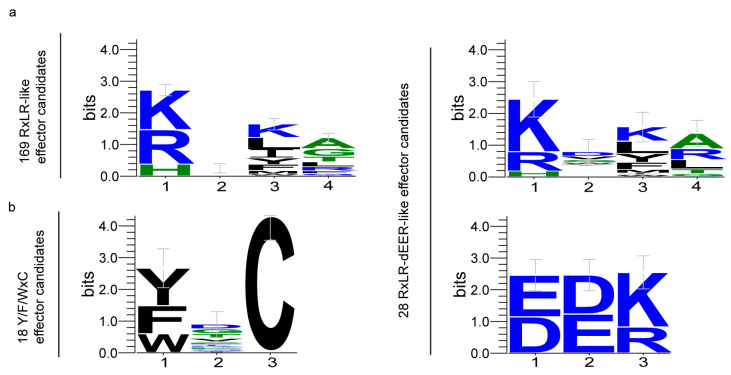
Hidden Markov models of conserved regions. (**a**) Weblog was used to analyze the hidden Markov models of conserved regions for 169 RXLR-like and 28 RXLR-deeR-like effector candidates. (**b**) Weblog was used to analyze the hidden Markov models of conserved regions for 18 Y/F/WxC effector candidates.

**Figure 3 ijms-25-00650-f003:**
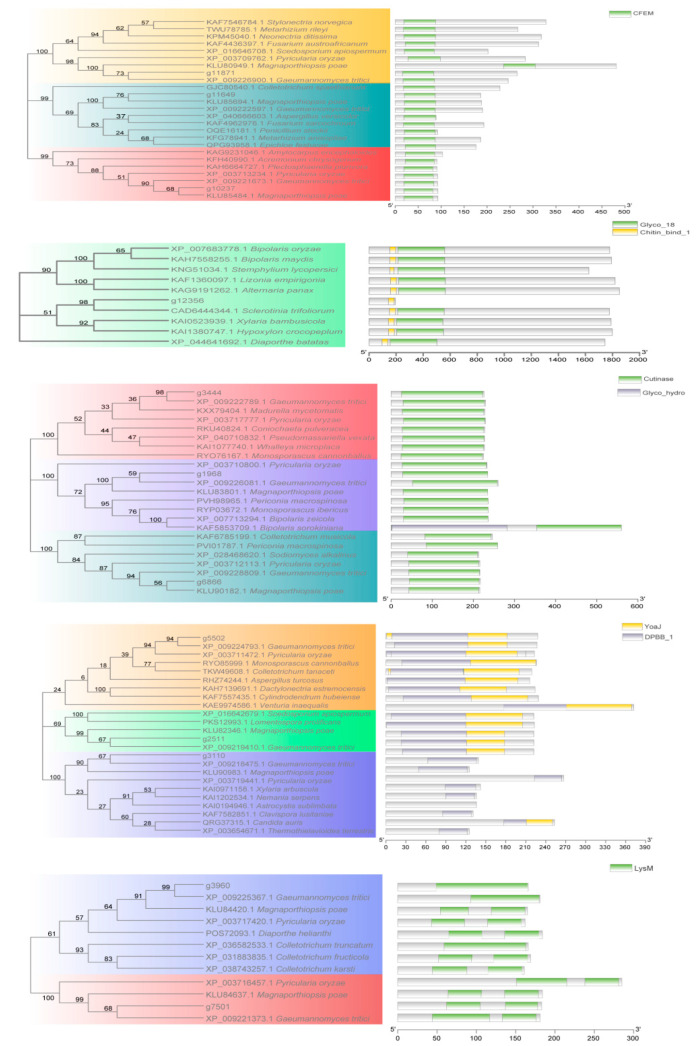
Phylogenetic analysis of effector candidates containing LysM, CFEM, DPBB_1, Cutinase, Chitin_bind domains, and other neighbor proteins. MEGA X analyzed all full-length sequences using an unrooted neighbor-joining bootstrap (1000 replicates).

**Figure 4 ijms-25-00650-f004:**
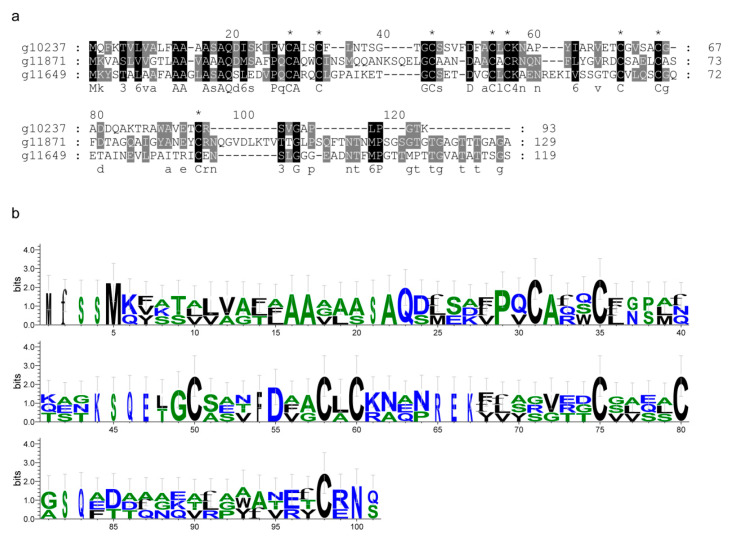
Identification of CFEM effector candidates in *F. oryzae*. (**a**) Multiple sequence alignment of CFEM domain-containing effector candidates in *F. oryzae*. The full sequences of the proteins were aligned by Clustal X. The darkness level of the black shadow indicates the amino acid composition at each position. * represents eight cysteine residues. (**b**) The hidden Markov models of conserved regions for CFEM effector candidates by Weblog.

**Figure 5 ijms-25-00650-f005:**
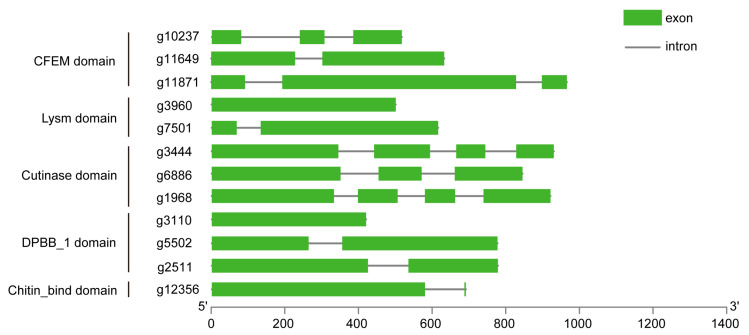
The arrangement and quantity of introns and exons. By comparing the DNA sequences and codon sequences, the identification of introns and exons was determined in terms of their position and number.

**Figure 6 ijms-25-00650-f006:**
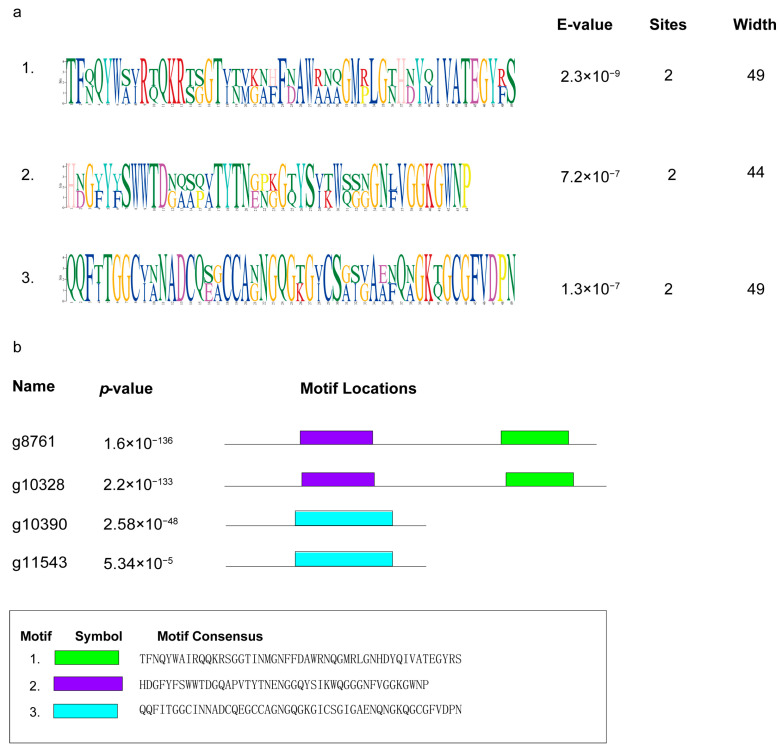
Three novel motifs of effector candidates by MEME. (**a**) The prediction of three novel motifs of effector candidates by MEME. (**b**) The location of three novel motifs in effector candidates.

**Figure 7 ijms-25-00650-f007:**
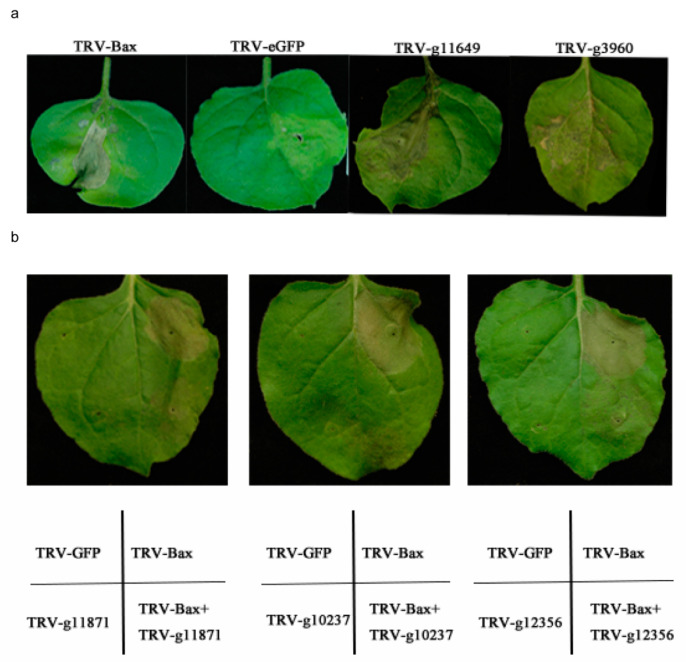
The transient expression of effector candidates in *N. benthamiana* leaves. (**a**) Effector candidates with conserved domains (LysM, CFEM, Cutinase, DPBB_1, and Chitin_bind) were constructed as transient expression vectors and introduced into N. benthamiana leaves through *A. tumefaciens* infiltration, with Bax serving as a positive control. (**b**) The effector candidates suppressed the cell death induced by Bax in *N. benthamiana* leaves. Comparison of *N. benthamiana* leaf images 5 days after co-infiltration with TRV-Bax, in contrast to TRV-eGFP used as a negative control.

**Figure 8 ijms-25-00650-f008:**
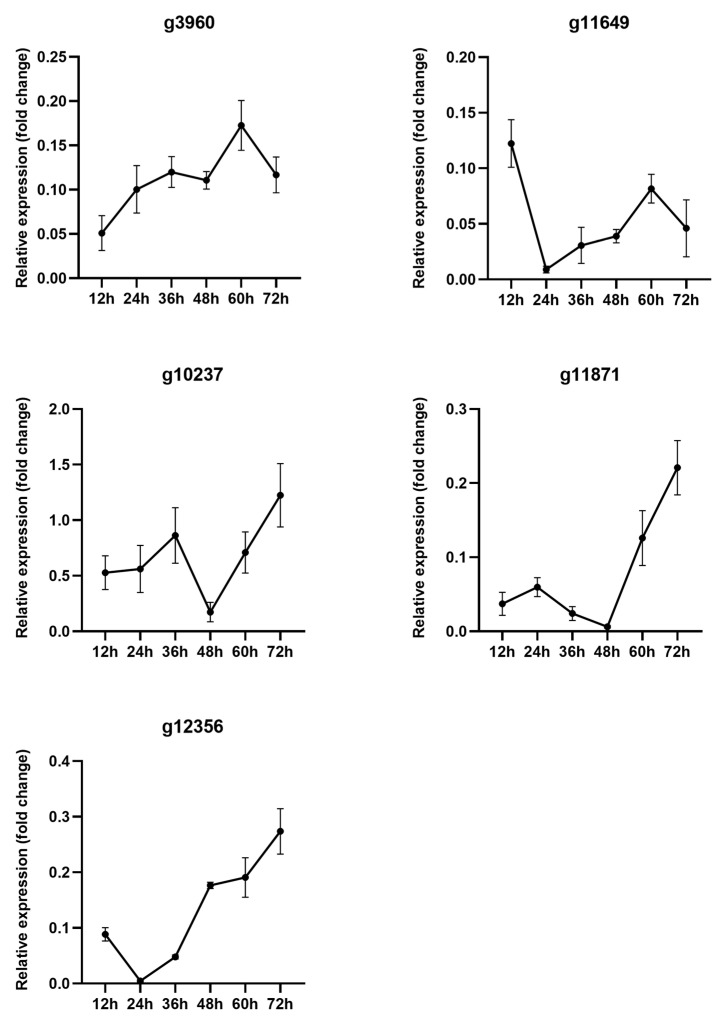
The relative expression level of effector candidates in the initial stage of infection. The gene’s relative expression is determined using the 2^−ΔCt^ method. The information signifies averages, and the error bars indicate the standard deviations obtained from six samples that are biological replicates.

**Table 1 ijms-25-00650-t001:** Genome-wide identification of effector candidates with CFEM, Cutinase, LysM, Chitin_bind, and DPBB_1 domains in *F.oryzae*.

Gene Accession	Conserved Domain (Pfam)	EffectorP Prediction	Effector Probability	WT FPKM	2 dpi FPKM	6 dpi FPKM	15dpi FPKM
g10237	CFEM domain (PF05730)	Apoplastic effector	0.829	12.1	1128.3	2355.9	1165.4
g11649	CFEM domain (PF05730)	Apoplastic effector	0.651	0.5	21.5	104.5	42.2
g11871	CFEM domain (PF05730)	Apoplastic effector	0.672	1.1	2.3	3.6	2.3
g3444	Cutinase (PF01083)	Apoplastic effector	0.826	3.8	18.5	139.2	53.9
g6886	Cutinase (PF01083)	Apoplastic effector	0.629	4.4	42.9	183.4	76.9
g1968	Cutinase (PF01083)	Apoplastic effector	0.667	1.7	182.6	474.5	219.6
g12356	Chitin_bind_1 (PF00187)	Cytoplasmic effector	0.843	0.9	4.6	5.4	3.6
g3110	DPBB_1 (PF03330)	Apoplastic effector	0.501	1.1	2.9	3.2	2.4
g5502	DPBB_1 (PF03330)	Apoplastic effector	0.789	1.2	8.5	12.5	7.4
g2511	DPBB_1 (PF03330)	Apoplastic effector	0.852	0.4	0.9	9.6	3.6
g3960	LysM domain (PF01476)	Apoplastic effector	0.896	1.9	143.2	99.3	81.4
g7501	LysM domain (PF01476)	Apoplastic effector	0.803	9.6	20.3	24.3	18.1

## Data Availability

The supporting dataset for the findings of this article is available in this article along with its Appendix A.

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
