# Peer review of "Genome-Wide Identification and Characterization of Effector Candidates with Conserved Motif in Falciphora oryzae"

_ijms, 2024, doi:10.3390/ijms25010650_

Round 1
Reviewer 1 Report
Comments and Suggestions for Authors
Genome-wide identification and characterization of effector candidates with conserved motif in dark septate endophyte Falciphora oryzae
Summary
The study presents data on an important research topic in plant-pathogen interactions. The authors have examined fungal effectors in Falciphora oryza using bioinformatics tools and searching for effectors in databases such as SignalP and EfectorP. These approaches are novel, and a strength of the paper was that the candidates were tested experimentally by cloning and qRT-PCR. The use of both database and experimental methods are strengths of the experimental design.
General concept comments
The title suggests that the authors sought to find several candidate genes, but in the objectives, they mention only one gene as their aim in line 21. The objective needs to be written clearly to reflect the title of the manuscript.
Generally, the introduction, methods and results are sound. One weakness seems to be in presenting the details of the methods and results at the end of the introduction in lines 81-85. The introduction should introduce the topic, provide the research aims, and indicate what is novel about the current study, highlighting its importance. The authors also emphasize dark septate endophytes in the title and sometimes refer to these fungi throughout the paper. Are dark septate endophytes more important than other endophytes? Can they clarify why this is important? If not, I suggest that you remove the phrase from the title.
Specific comments
1. Line 21, page 1, “the aims to discover the gene…” This is not entirely accurate based on the title of the paper. Please rephrase and write more clearly.
2. Line 153, page 4. Motifs with high e-values? Did you mean low e-values? Any e-value over 0.05 is not significant. MEME Results (meme-suite.org)
3. On page 7 and Figure 5, some genes are not present in Table 1. For example, g5318, g9581, g12899 and g613 . Line 148, Page 4
4. Many references are more than 10 years old; the authors should seek to use more recent references that review new information on fungal effectors.
5. I have highlighted several typos and grammatical errors, but the authors must check these thoroughly. For example, the LysM domain is not written consistently throughout the manuscript.

I have highlighted several typos and grammatical errors, but the authors must check these thoroughly. For example, the LysM domain is not written consistently throughout the manuscript.
Reviewer 2 Report
Comments and Suggestions for Authors
The research explores an important and relevant topic, highlighting its significance in the field. The methods are well-explained, demonstrating a clear understanding of the chosen techniques and their application. The execution of the methodology appears rigorous and appropriately applied. However, while the introduction effectively introduces the research topic, it could benefit from better contextualization. Some parts contain short phrases with disconnected ideas, which, when further elaborated, could offer a more cohesive narrative. As an example, in line 45, the authors start by discussing the symbiotic relationship between rice and F. oryzae and its potential as a biocontrol agent, but it then shifts to discussing the importance of effectors as a virulence characteristic, without providing any context (strangely enough, there is better contextualization in the abstract). However, my bigger concern is the discussion section, that falls short of providing an in-depth analysis. It lacks a comprehensive discussion and seems more descriptive than analytical. It would greatly benefit from a deeper exploration and interpretation of the results.
In summary, the research paper is founded on an important research topic, and the methodology is well-explained and correctly applied. However, improvements in contextualizing the introduction, enriching the discussion, and enhancing the interpretation of results would further strengthen the paper’s overall contribution to the field.
Minor comments:
I believe in lines 38, 72 and 73, it would be better to reference the team and not just the first author, i.e., Yuan et al., instead of Yuan.
Round 2
Reviewer 2 Report
Comments and Suggestions for Authors
The authors have shown remarkable dedication and commitment in refining the manuscript, resulting in a considerable improvement in its overall quality. The contextualization in the introduction has been notably enhanced, providing a clearer background for the study. Additionally, the discussion section reflects a significant improvement, contributing to a more robust and comprehensive understanding of the findings. With these enhancements, I believe the manuscript is ready for publication.